# Reconstituted cell-free protein synthesis using in vitro transcribed tRNAs

Keita Hibi[1], Kazuaki Amikura [1,2], Naoki Sugiura [1], Keiko Masuda[3], Satoshi Ohno[4], Takashi Yokogawa [4], Takuya Ueda[1,5] & Yoshihiro Shimizu [3✉]

Entire reconstitution of tRNAs for active protein production in a cell-free system brings flexibility into the genetic code engineering. It can also contribute to the field of cell-free synthetic biology, which aims to construct self-replicable artificial cells. Herein, we developed a system equipped only with in vitro transcribed tRNA (iVTtRNA) based on a reconstituted cell-free protein synthesis (PURE) system. The developed system, consisting of 21 iVTtRNAs without nucleotide modifications, is able to synthesize active proteins according to the redesigned genetic code. Manipulation of iVTtRNA composition in the system enabled genetic code rewriting. Introduction of modified nucleotides into specific iVTtRNAs demonstrated to be effective for both protein yield and decoding fidelity, where the production yield of DHFR reached about 40% of the reaction with native tRNA at 30°C. The developed system will prove useful for studying decoding processes, and may be employed in genetic code and protein engineering applications.

[1] Department of Computational Biology and Medical Sciences, Graduate School of Frontier Sciences, The University of Tokyo, Kashiwa, Chiba 277-8562, Japan. [2] Department of Molecular Biophysics and Biochemistry, Yale University, New Haven, CT 06520, USA. [3] Laboratory for Cell-Free Protein Synthesis, RIKEN Center for Biosystems Dynamics Research (BDR), Suita, Osaka 565-0874, Japan. [4] Department of Chemistry and Biomolecular Science, Faculty of Engineering, Gifu University, Gifu 501-1193, Japan. [5] Department of Integrative Bioscience and Biomedical Engineering, Graduate School of Science and Engineering, Waseda University, Tokyo, Shinjuku 162-8480, Japan. ✉email: yshimizu@riken.jp

Transfer RNAs (tRNAs) are key molecules that carry amino acids to ribosomes and decode the genetic code using anticodons to translate genetic information into the primary sequence of a polypeptide chain. Since the first pivotal study[1], tRNAs have been engineered to alter connections between the genetic code and amino acids and reassign the genetic code for site-specific incorporation of noncanonical amino acids into proteins in a cell-free protein synthesis system[2,3]. Along with the development of aminoacyl-tRNA synthetase (aaRS) engineering and ribozyme utilization via flexizymes[3], the creation of novel functional peptides or proteins with noncanonical amino acids has been extensively examined. Such genetic code engineering is expected to be highly beneficial for drug development research[4].

Despite considerable effort expended on genetic code engineering, complete reconstitution of tRNAs in a cell-free system, which could prove game-changing for codon reassignment, has not been fully realized to our knowledge. The major obstacle to achieving this goal is the presence of a variety of modified nucleotides in tRNA structures. In contrast with many non-essential tRNA modifications outside anticodons[5], those in the anticodon loop region directly interact with the mRNA codon on the ribosome, and become essential in some cases. Among the most widely-studied examples are modifications in tRNA$^{Lys}_{UUU}$. The anticodon loop of tRNA$^{Lys}_{UUU}$ is modified at position 34 (5-methylaminometyl-2-thiouridine; mnm$^5$s$^2$U34) and 37 (N6-threonylcarbamoyladenosine; t$^6$A37). These modifications are essential in decoding processes by both biochemical and structural analyses[6,7]. Nevertheless, several researchers have tried to entirely reconstitute the tRNA in a cell-free protein expression system utilizing a set of in vitro transcribed tRNAs (iVTtRNAs). Synthesizing active proteins using a set of three native tRNAs and 48 iVTtRNAs[8] or redesigning the genetic code using a set of 32 iVTtRNAs to incorporate noncanonical amino acids into synthesized polypeptides[9] have been studied so far. Chemically synthesized 21 tRNAs have been used for the in vitro protein synthesis experiment though detailed analysis of individual effect of each added tRNA was not examined[10].

Notably, such a research direction can also contribute to the field of cell-free synthetic biology, which aims to construct artificial cells based on encapsulation of cell-free systems in a synthetic compartment such as phospholipid vesicles or microfluidic chips[11,12]. One of the major goals of this field is the construction of a self-replicable artificial cell by organizing essential purified biological macromolecules[13,14]. Such bottom-up approaches can be used for the construction of minimal cells, whereas top-down approaches utilize natural living cells possessing entirely synthesized genomes[15,16]. This may appear to be a formidable task requiring the identification, preparation, and integration of essential molecules for reconstituting self-replication. Nevertheless, it could be useful to explore the boundary between living and nonliving systems because self-replication is one of the most fundamental features of life.

The PURE system is a reconstituted cell-free protein synthesis system composed of individually prepared components required for gene expression in Escherichia coli[17]. This system is suitable for accomplishing self-replicable artificial cells via a bottom-up approach[18]. As mentioned in a previous review[13], exploring a set of macromolecules that can create a new automaton composed of the same molecules, the basis of von Neumann's theory of self-replication, is crucial for the construction of self-reproducing systems. Thus, in the present study, we focused on the reconstitution of transfer RNA (tRNA) in a controllable manner to explore a minimal set of tRNAs that are functional for protein expression within the PURE system, and that can be synthesized by the PURE system.

Herein, we present a cell-free protein synthesis system, based on the PURE system, equipped only with iVTtRNAs. The system, consisting of a set of 21 iVTtRNAs without any modifications, is able to synthesize active proteins according to the redesigned simplified genetic code. We succeeded in producing active dihydrofolic acid reductase (DHFR) and super-folder green fluorescent protein (sfGFP) variants with comparable activity to proteins synthesized using native tRNA mixtures. We also successfully rewrote the genetic code by assigning Ala to the Ser codon (UCG) through engineering tRNA$^{Ala}_{GGC}$. The developed system displayed orthogonality in terms of decoding and produced active DHFR according to the redesigned genetic code, whereas inactive polypeptides were synthesized using a native tRNA mixture based on the universal genetic code. Introduction of modified nucleotides into specific iVTtRNAs further demonstrated to be effective for both protein yield and decoding fidelity.

## Results

**Selection and preparation of iVTtRNAs**. E. coli tRNAs contain a number of modified nucleotides, among which modified nucleotides 34 and 37 are essential for decoding specific codons[19]. In addition, in E. coli, some genes responsible for the modification of these nucleotides are essential genes[20,21]. Therefore, reconstitution of protein synthesis using a full set of iVTtRNAs may not be practical. However, codons are degenerate, and most amino acids are encoded by multiple codons. Therefore, by using a restricted set of tRNAs that may not require these modified nucleotides, reconstitution of tRNAs in the PURE system may be possible.

Nucleotide 34 is directly involved in base pairing at the wobble position, and can be modified in various ways (Fig. 1a and Supplementary Fig. 1). In particular, when the nucleotide U is present, it tends to be modified. Therefore, we preferentially selected tRNAs with a G or C at position 34, similar to a previous study[9]. However, Lys and Glu codons are decoded only by tRNA$^{Lys}_{UUU}$ and tRNA$^{Glu}_{UUC}$, respectively, in E. coli, in which nucleotide 34 is U and modified to mnm$^5$s$^2$U. Because these modifications are reportedly required for efficient aminoacylation[22,23], and are also essential for codon-anticodon interactions and subsequent translocation on the ribosome[6,7], it is predictable that the corresponding iVTtRNA may not be functional in the translation machinery. Therefore, we artificially prepared tRNA$^{Lys}_{CUU}$ and tRNA$^{Glu}_{CUC}$ in an attempt to improve codon-anticodon interactions on the ribosome, and the aminoacylation efficiency, as reported in a previous study[24]. When the anticodon sequence (positions 34−36) of tRNA is GUN (tRNA$^{Tyr}_{GUA}$, tRNA$^{His}_{GUG}$, tRNA$^{Asn}_{GUU}$, and tRNA$^{Asp}_{GUC}$), where N is any nucleotide, G at position 34 is modified to queuosine (Q). In such cases, we decided to use an unmodified version. It was reported that these iVTtRNAs can be aminoacylated without the modification[25−27]. Consequently, we selected 21 tRNAs for the reconstitution of tRNAs with iVTtRNAs in the PURE system from E. coli genomic sequence (Fig. 1b, colored yellow and blue in Supplementary Fig. 1, and Supplementary Data 1). Except for Met, in which both initiator and elongator tRNAs were selected, a single tRNA was selected for the decoding of each amino acid.

Preparation of iVTtRNAs was performed by separate runoff transcription reactions with T7 RNA polymerase. To prevent addition of an extra nucleotide to the 3′-terminus of tRNA, 2′-methoxy modification was introduced at the second nucleotide from the 5′-terminus of the antisense strand in PCR-amplified DNA templates using modified DNA primers[28]. We also followed a previous protocol using RNase P for the efficient transcription[29]. Precursor tRNA with 27 extra nucleotides universally

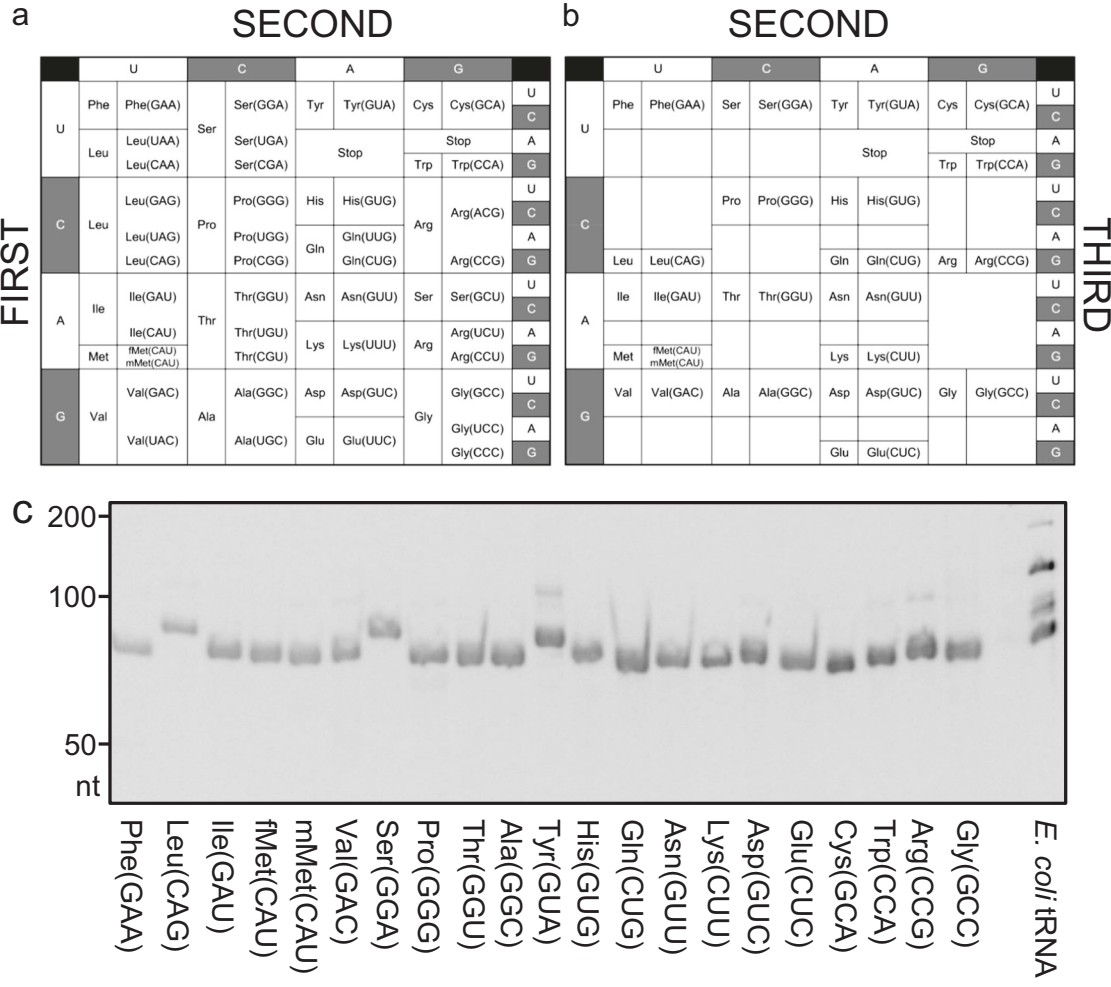

**Fig. 1 Selection and preparation of iVTtRNAs for tRNA reconstitution.** Genetic code table showing 41 tRNA species in *E. coli* (**a**) and the table showing 21 tRNAs used in this study (**b**) are shown. **c** PAGE analysis of the prepared 21 iVTtRNAs. Purified iVTtRNAs (0.01 A$_{260}$ units) and native tRNA mixtures (0.04 A$_{260}$ units) were analyzed by urea-PAGE.

introduced at the 5′-terminus was transcribed and processed using RNase P to generate mature tRNA, followed by purification with anion-exchange chromatography. All iVTtRNAs were successfully prepared and found to be uniform (Fig. 1c).

**Aminoacylation of iVTtRNAs.** The ability of the prepared iVTtRNAs to accept amino acids was investigated by aminoacylation assays using each aaRS and radioisotope-labeled amino acids. All possible combinations of iVTtRNAs and aaRSs were tested to investigate orthogonality between aaRSs and iVTtRNAs (Supplementary Fig. 2). The measurements essentially demonstrated orthogonality, even in the absence of any modifications (i.e., only the amino acid corresponding to each iVTtRNA was specifically aminoacylated). The ratios of amino acid acceptance for each tRNA were also sufficient to support translation (15–50%), except for iVTtRNA$^{Ile}_{GAU}$ (4.2%) and iVTtRNA$^{Glu}_{CUC}$ (0.5%). The acceptance ratio for iVTtRNA$^{Glu}_{CUC}$ was particularly low, and this made it difficult to confirm orthogonality (Supplementary Fig. 2p).

These observations are consistent with previous studies in which threonylcarbamoyladenosine (t$^6$A) at position 37 in tRNA$^{Ile}_{GAU}$ and mnm$^5$s$^2$U at position 34 in tRNA$^{Glu}_{UUC}$ were shown to be important for aminoacylation efficiency[30,31]. Therefore, we tried to compensate for the lack of modification

in these tRNAs by increasing the aaRS concentration. By increasing the isoleucyl-tRNA synthetase and glutamyl-tRNA synthetase concentration 30-fold (from 50 nM to 1.5 μM), the amino acid acceptance ratios were increased to 15% (isoleucylation) and 13% (glutamylation), with sufficient orthogonality (Supplementary Fig. 3). The results indicate that all of the prepared iVTtRNAs could be sufficiently aminoacylated by each aaRS and were therefore suitable for construction of an iVTtRNA-based cell-free protein synthesis system.

**Decoding of iVTtRNAs.** We next investigated which codons could be decoded by the prepared iVTtRNAs in the PURE system. Hypothetically, decoding fidelity is maintained by canonical Watson-Click base pairing, while the third base pair in the codon-anticodon interaction is less stringent and can be "wobble." Although there are some exceptions such as the stop codon (UAG and UAA) decoded by Gln-tRNA$^{Gln}_{CUG}$[32] and the GGC codon (Gly) decoded by Ser-tRNA$^{Ser}_{GCU}$[33], the efficiency of such unusual decoding is very low, hence we investigated all four codons in which the first and second base pairs engage in canonical Watson–Crick base pairing for each iVTtRNA.

Assays were performed by using template DNA including test codons that encode octapeptide (Supplementary Data 2), and peptide synthesis in the PURE system containing reconstituted

iVTtRNAs was evaluated by measuring radioisotope-labeled amino acid incorporation into the trichloroacetic acid (TCA)-insoluble fraction. Hydrophobic amino acids such as Leu, Phe, Ile, and Val were used as components of the synthesized peptides to ensure the insolubility. At first, the peptide sequence MFFLFXLF, where X is the test amino acid to be evaluated, was used for evaluation of iVTtRNA$^{Ile}_{GAU}$, iVTtRNA$^{Val}_{GAC}$, iVTtRNA$^{mMet}_{CAU}$, and iVTtRNA$^{fMet}_{CAU}$. This sequence was derived from our previous manuscript[34] where decoding abilities of iVTtRNA$^{Phe}_{GAA}$ and iVTtRNA$^{Leu}_{CAG}$ have already been confirmed. Next, we switched the sequence from MFFLFXLF to MIIIIXLF to increase the hydrophobicity of synthesized peptides because testing of some hydrophilic amino acids were found to be difficult by using the sequence MFFLFXLF. After examining other iVTtRNAs except for iVTtRNA$^{Phe}_{GAA}$ and iVTtRNA$^{Leu}_{CAG}$, the sequence MIIIIXVV was used to evaluate iVTtRNA$^{Phe}_{GAA}$ and iVTtRNA$^{Leu}_{CAG}$. To investigate the decoding ability of each iVTtRNA in a highly sensitive manner, each iVTtRNA was added in excess relative to the ribosome (6 μM of each tRNA, 0.2 μM ribosome).

The results were both similar and different to those of previous studies[8,9] (Supplementary Figs. 4–8). Each iVTtRNA with an anticodon starting with the nucleotide G (iVTtRNA$^{Phe}_{GAA}$, iVTtRNA$^{Ile}_{GAU}$, iVTtRNA$^{Val}_{GAC}$, iVTtRNA$^{Ser}_{GGA}$, iVTtRNA$^{Pro}_{GGG}$, iVTtRNA$^{Thr}_{GGU}$, iVTtRNA$^{Ala}_{GGC}$, iVTtRNA$^{Tyr}_{Tyr}$, iVTtRNA$^{His}_{GUG}$, iVTtRNA$^{Asn}_{GUU}$, iVTtRNA$^{Asp}_{GUC}$, iVTtRNA$^{Cys}_{GCA}$, and iVTtRNA$^{Glys}_{GCC}$) effectively decoded target codons ending with the nucleotides U or C. Presumably due to the high concentration of iVTtRNA, iVTtRNA$^{Ile}_{GAU}$, and iVTtRNA$^{Asn}_{GUU}$, which have been demonstrated not to decode target codons in previous studies, were shown to be functional. In addition, iVTtRNA$^{Tyr}_{GUA}$, iVTtRNA$^{His}_{GUG}$, iVTtRNA$^{Asn}_{GUU}$, and iVTtRNA$^{Asp}_{GUC}$, that have Q at 34 position in their native state (Q series iVTtRNAs), decoded both target codons, while they tended to decode codons ending in C better than those ending in U.

All iVTtRNAs for which their anticodons started with the nucleotide C (iVTtRNA$^{Leu}_{CAG}$, iVTtRNA$^{mMet}_{CAU}$ and/or iVTtRNA$^{fMet}_{CAU}$, iVTtRNA$^{Gln}_{CUG}$, iVTtRNA$^{Lys}_{CUU}$, iVTtRNA$^{Glu}_{CUC}$, iVTtRNA$^{Trp}_{CCA}$, and iVTtRNA$^{Arg}_{CCG}$) effectively decoded target codons ending with the nucleotide G. These iVTtRNAs were divided into two types; iVTtRNA$^{Gln}_{CUG}$, iVTtRNA$^{Lys}_{CUU}$, iVTtRNA$^{Trp}_{CCA}$, and iVTtRNA$^{Arg}_{CCG}$ that only decoded codons ending with G, and iVTtRNA$^{Leu}_{CAG}$ and iVTtRNA$^{Glu}_{CUC}$ that also decoded codons ending in A. The former group is consistent with a previous study[8], and the latter might reflect the addition of excess iVTtRNAs in the cell-free expression system. We note that the buffer composition was based on potassium glutamate, which is known to be an efficient ingredient for the cell-free protein synthesis reaction[35], and glutamic acid was present in the system in submolar quantities. This may cause efficient decoding with unmodified iVTtRNA$^{Glu}_{CUC}$ in addition to excess addition of iVTtRNA$^{Glu}_{CUC}$. Further information on the detailed differences between decoding properties among iVTtRNAs and inconsistencies with the genetic code table can be found in Supplementary Note 1, but the results presented here indicate the possibility of synthesizing proteins using a set of iVTtRNAs without any native tRNA.

**Cell-free protein synthesis using iVTtRNAs**. To evaluate the activity of iVTtRNAs for the synthesis of long polypeptides (i.e., protein synthesis), DHFR and sfGFP synthesis were examined using the PURE system containing iVTtRNAs. DNA templates for protein expression (Supplementary Data 3) were prepared according to the genetic code table where each amino acid corresponds to only a single codon (Fig. 2a). To decrease the GC content in order to avoid internal secondary structure formation

within mRNAs, the use of wobble base pairings were prioritized for iVTtRNAs whose anticodons start with the nucleotide G, while they were not used for Q series iVTtRNAs that may affect the decoding efficiency.

Despite lacking any modifications, iVTtRNAs were shown to support the synthesis of both proteins, and products synthesized using iVTtRNAs had the same molecular weight as those synthesized using native tRNA mixtures (Fig. 2b, c). Two temperatures were tested, and the yields of proteins synthesized using iVTtRNAs were higher when incubated at 30 °C than 37 °C, whereas yields with native tRNA mixtures were higher at 37 °C than at 30°C (Fig. 2d, e, and Supplementary Fig. 9). We note that sfGFP synthesis was also successful without using any wobble base pairings (Supplementary Fig. 10 and Supplementary Data 3).

The relationship between the productivity of DHFR synthesis and tRNA concentration was monitored, and it increased linearly with increasing iVTtRNAs added, while it was saturated when 60 A$_{260}$ unit/mL native tRNA mixtures were added (Supplementary Fig. 11). These observations suggest the possibility that some reactions involving inefficient iVTtRNAs, such as aminoacylation and decoding, may become rate-limiting reactions, and do not reach a plateau when iVTtRNAs are used. Therefore, we fixed the iVTtRNA concentration at 20 A$_{260}$ unit/mL and excessively added only four iVTtRNA species (iVTtRNA$^{Ile}_{GAU}$, iVTtRNA$^{Pro}_{GGG}$, iVTtRNA$^{Asn}_{GUU}$, and iVTtRNA$^{Glu}_{CUC}$), which appeared to be inefficient from aminoacylation experiments in this study (Supplementary Figs. 2 and 3) and previous in vitro tRNA reconstitution studies[8,9]. The results demonstrated that higher yields of DHFR were obtained when specific iVTtRNAs were added, compared with a mixture of all iVTtRNAs, suggesting that reactions involving these specific iVTtRNAs are rate-limiting (Fig. 3a).

The addition of specific iVTtRNAs was also shown to affect the decoding fidelity. The specific activity of synthesized DHFR was measured, and when the protein was synthesized using a mixture of equally abundant iVTtRNAs, the activity was almost half of the protein synthesized using native tRNA mixtures (Fig. 3b). By contrast, activity was recovered to levels comparable with the protein synthesized using native tRNAs when specific iVTtRNAs were added (Fig. 3b). A previous study showed that tRNAs without modifications in anticodon loops can mistakenly decode near-cognate codons[36]. Similarly, our results suggest that a decrease in decoding fidelity may be recovered by manipulating the tRNA composition, specifically by increasing the concentrations of inefficient tRNAs.

**Genetic code redesigning using iVTtRNAs**. We explored the possibility of redesigning the genetic code table using iVTtRNAs. It is known that alanyl-tRNA synthetase (AlaRS) specifically recognizes the G2:U71 wobble base pair in the acceptor stem region of tRNA$^{Ala}$, and this base pair is the major identity element that allows AlaRS to discriminate tRNA$^{Ala}$ from other tRNAs[37]. By contrast, tRNA$^{Ser}$ has a long variable loop as its identity element for discrimination by seryl-tRNA synthetase[27]. Both aaRS do not recognize anticodon regions for the discrimination and thus, we tried to reassign Ala to the Ser codon by transplanting the CGA anticodon into tRNA$^{Ala}_{GGC}$, to generate tRNA$^{Ala}$ with a CGA anticodon (iVTtRNA$^{Ala}_{CGA}$), corresponding to a Ser UCG codon (Supplementary Fig. 12). The scheme utilizing the identity element for AlaRS has been adopted for genetic code engineering studies[38].

After preparing an appropriate DNA template (Supplementary Data 3) for DHFR based on the newly designed genetic code (Fig. 4a), in which all GCU codons used in experiments to generate the results shown in Fig. 2 were overwritten by the UCG

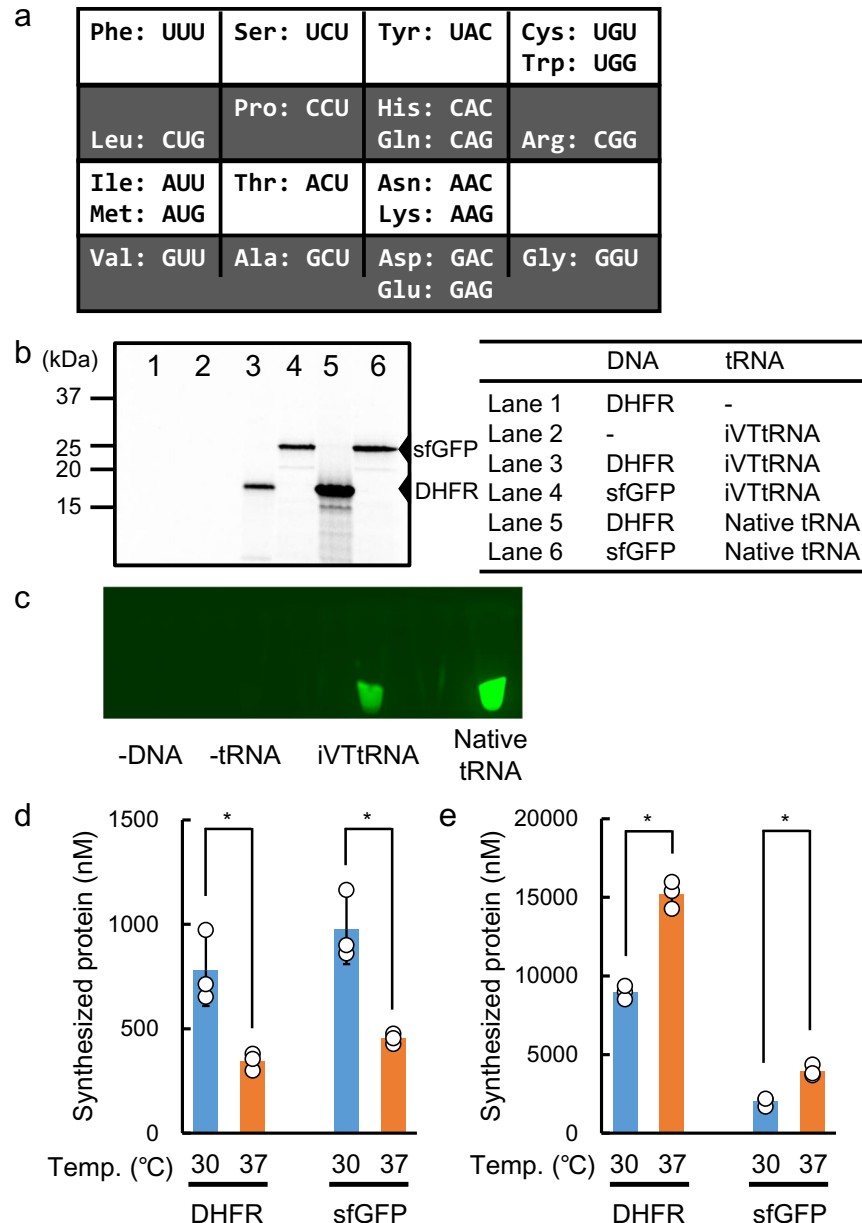

**Fig. 2 Protein synthesis with iVTtRNAs. a** Genetic code table used for protein synthesis with iVTtRNAs. DNA sequences for protein expression were designed according to this table. **b** SDS-PAGE analysis of synthesized proteins labeled with [$^{35}$S]Met. **c** Fluorescence images of synthesized sfGFP in reaction mixtures. Temperature dependence of the synthesized protein yield. Yields with iVTtRNAs (**d**) and yields with native tRNA mixtures (**e**) are shown. All experiments were performed with 60 A$_{260}$ unit/mL iVTtRNA mixtures or 40 A$_{260}$ unit/mL native tRNA mixtures. Error bars indicate standard deviation of independent repeats of triplicate measurements. Each dot represents individual observed value. An asterisks indicate that P values < 0.05. Welch's t test was applied between yields at 30 and 37 °C.

codon, DHFR synthesis was performed with iVTtRNAs complemented with iVTtRNA$^{Ala}_{CGA}$ or native tRNA mixtures. Of note, excessive addition of the four specific iVTtRNAs used in experiments that generated the results shown in Fig. 3 were also performed in this experiment. Amino acid sequences of the synthesized DHFR were hypothetically different between two conditions; the correct amino acid sequence should only be synthesized using the new iVTtRNA mixtures, whereas all Ala residues should be changed to Ser when native tRNA mixtures are used, according to the canonical genetic code. SDS-PAGE analysis of the synthesized DHFR revealed a slight band shift, indicating that different polypeptides were produced in the two conditions (Fig. 4b). Liquid chromatography-mass spectrometry (LC-MS)

analysis confirmed the successful codon reassignment (Supplementary Fig. 13 and Supplementary Data 4). Specific activity measurements of synthesized DHFR clearly demonstrated that only DHFR synthesized using iVTtRNAs possessed activity comparable with DHFR harboring the correct amino acid sequence (compare Fig. 3b with Fig. 4c), whereas DHFR synthesized using native tRNA mixtures was inactive. These results indicate that Ala was not incorporated into DHFR when native tRNA mixtures were used, whereas the UCG codon was successfully reassigned to Ala when iVTtRNAs were employed.

**Introduction of modified nucleotides into iVTtRNAs.** Excessive addition of four iVTtRNA species (iVTtRNA$^{Ile}_{GAU}$,

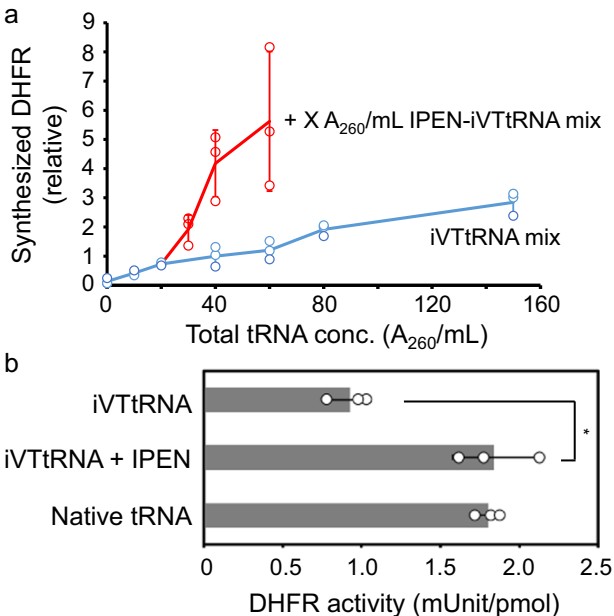

**Fig. 3 Effects of addition of four specific iVTtRNAs. a** Dependency of tRNA concentration on the yield of synthesized DHFR. The red line shows the results of added all iVTtRNAs, while the blue line shows the results of adding 20 $A_{260}$ unit/mL iVTtRNAs with a specified amount of the IPEN-iVTtRNA mixture composed of four specific iVTtRNAs (iVTtRNA$^{Ile}_{GAU}$, iVTtRNA$^{Pro}_{GGG}$, iVTtRNA$^{Glu}_{CUC}$, and iVTtRNA$^{Asn}_{GUU}$). Experiments were performed with 0, 10, 20, and 40 $A_{260}$ unit/mL IPEN-iVTtRNA mix. **b** Specific activity of synthesized DHFR. Activities are shown for synthesized DHFR prepared with 60 $A_{260}$ unit/mL iVTtRNA mixtures, 20 $A_{260}$ unit/mL iVTtRNA mixtures, and 40 $A_{260}$ unit/mL IPEN-iVTtRNA mixtures, and 40 $A_{260}$ unit/mL native tRNA mixtures. Error bars indicate standard deviation of independent repeats of triplicate measurements. Each dot represents individual observed value. An asterisk indicates that $P$ value < 0.05. Welch's $t$ test was applied between DHFR activities synthesized with iVTtRNA mixtures and iVTtRNA and IPEN-iVTtRNA mixtures.

iVTtRNA$^{Pro}_{GGG}$, iVTtRNA$^{Asn}_{GUU}$, and iVTtRNA$^{Glu}_{CUC}$) were shown to be effective for both yields and fidelity (Fig. 3). Therefore, we addressed this matter by introducing modification on anticodon loop regions of these iVTtRNA. t6A modification at position 37 of iVTtRNA$^{Ile}_{GAU}$ and iVTtRNA$^{Asn}_{GUU}$, 1-methylguanosine (m1G) modification at position 37 of iVTtRNA$^{Pro}_{GGG}$, and 5-methylaminomethyluridine (mnm5U) modification at position 34 of iVTtRNA$^{Glu}_{UUC}$ were successfully introduced using corresponding modification enzymes (Supplementary Fig. 14). We note that tRNA$^{Glu}_{UUC}$ that has a native anticodon sequence was used in this experiment instead of tRNA$^{Glu}_{CUC}$.

Aminoacylation experiments showed that the modification on iVTtRNA$^{Ile}_{GAU}$, iVTtRNA$^{Pro}_{GGG}$, and iVTtRNA$^{Glu}_{UUC}$ were effective for increasing the aminoacylation efficiency whereas that for iVTtRNA$^{Asn}_{GUU}$ did not show improvement (Supplementary Fig. 15). Particularly, t6A37 modification on iVTtRNA$^{Ile}_{GAU}$ showed about sevenfold increase in the efficiency while mnm5U34 modification on iVTtRNA$^{Glu}_{UUC}$ and m1G37 modification on iVTtRNA$^{Pro}_{GGG}$ showed about 2- and 1.4-fold increase, respectively. We next examined the DHFR synthesis using modified or unmodified iVTtRNA$^{Ile}_{GAU}$, iVTtRNA$^{Pro}_{GGG}$, and iVTtRNA-$^{Glu}_{UUC}$ with all possible combinations. The result demonstrated the addition of modified iVTtRNA$^{Ile}_{GAU}$ increased the yield and the addition of modified iVTtRNA$^{Glu}_{UUC}$ further increased the

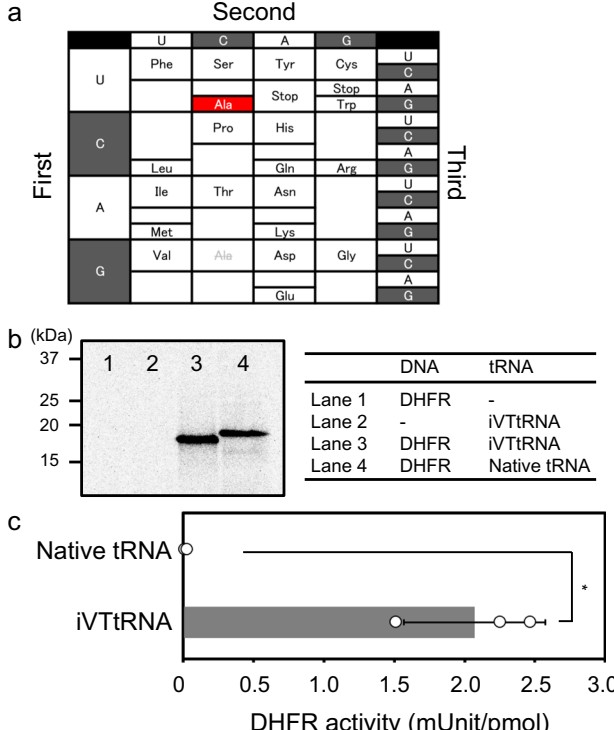

**Fig. 4 Genetic code redesigning with iVTtRNAs. a** Redesigned genetic code table using newly prepared iVTtRNA$^{Ala}_{CGA}$ but without iVTtRNA$^{Ala}_{GGC}$. The newly assigned UCG codon for Ala is highlighted in red, and original Ala codons not assigned to any amino acids in the redesigned genetic code table are shaded with gray. **b** SDS-PAGE analysis of synthesized protein labeled with [35S]Met. **c** Specific activity of synthesized DHFR. All experiments were performed with 40 $A_{260}$ unit/mL native tRNA mixtures or 20 $A_{260}$ unit/mL iVTtRNA mixtures and 40 $A_{260}$ unit/mL IPEN-iVTtRNA mixtures. Error bars indicate standard deviation of independent repeats of triplicate measurements. Each dot represents individual observed value. An asterisk indicates that $P$ value < 0.05. Welch's $t$ test was applied between DHFR activities synthesized with native tRNA mixtures and iVTtRNA mixtures.

yield where it reached about 40% compared with the native tRNA mixtures (Fig. 5a, b). Specific activities of synthesized DHFR showed that the addition of modified iVTtRNA$^{Ile}_{GAU}$ showed the level comparable with DHFR synthesized with native tRNAs, suggesting t6A modification on iVTtRNA$^{Ile}_{GAU}$ is crucial for the decoding fidelity (Fig. 5c).

## Discussion

Herein, we demonstrated that the PURE system containing iVTtRNAs without any modified nucleotides is capable of synthesizing active proteins. The set of 21 iVTtRNAs (Fig. 1 and Supplementary Fig. 1) used in this study can be regarded as hypothetically minimal, and the resulting protein expression system based on a simplified genetic code table in which only a single tRNA corresponds to a single amino acid, including formyl-methionine, proved effective (Fig. 2a). It is interesting to note that exploring better set of the tRNA sequences which can improve the translation efficiency, which should performed with the developed system in the future. Natural genomic sequences cannot be used for the gene expression in this system and the gene synthesis technologies are required. We consider that such a feature does not become a limitation because of the

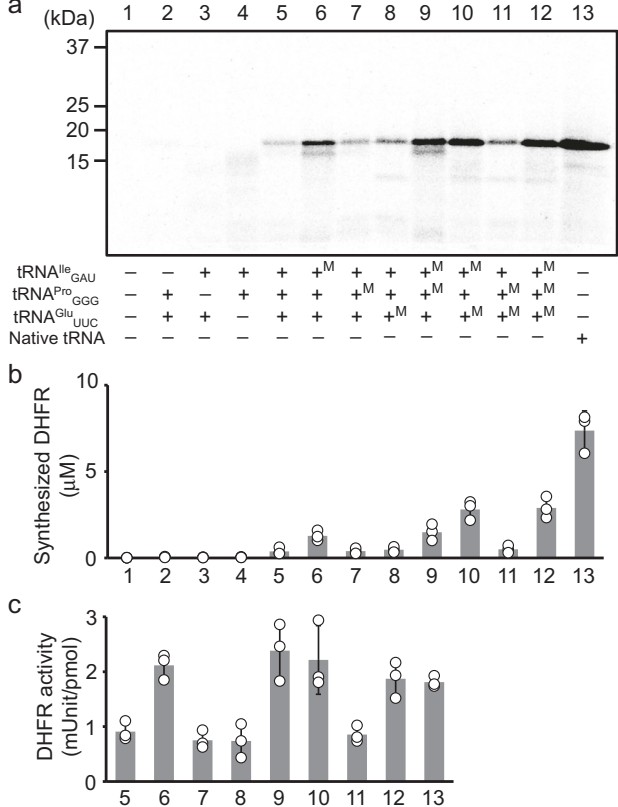

**Fig. 5 Effects of modifications on specific iVTtRNAs on protein synthesis yield and fidelity. a** SDS-PAGE analysis of synthesized proteins labeled with [$^{35}$S]Met. **b** Yields of synthesized DHFR. **c** Specific activity of synthesized DHFR. All experiments were performed with 40 A$_{260}$ unit/mL native tRNA mixtures or 60 A$_{260}$ unit/mL iVTtRNA mixtures. Error bars indicate standard deviation of independent repeats of triplicate measurements. Each dot represents individual observed value.

recent development of these technologies[39], which enables an easy access to the custom-designed DNA templates. When the composition of the iVTtRNAs was optimized, the activity of the synthesized protein was comparable to that synthesized with native tRNA mixtures (Fig. 3b), suggesting this set of 21 iVTtRNAs is intrinsically capable of maintaining translational fidelity. This may imply that primordial features of tRNAs, which may have functioned without any modifications when the genetic code first evolved, are still preserved in these selected tRNAs.

Extant tRNAs have a variety of modified nucleotides, and they are added through complicated enzymatic networks. For example, to introduce 2-thiourigine, the intermediate of mnm$^5$s$^2$U, an enzymatic network for sulfur transfer composed of seven proteins is necessary[40]. However, upon the appearance of the first translation apparatus, tRNAs may have performed their functions without any modifications, which require such complicated enzymatic networks. Thus, the present results might reflect the translation system during the early stages of life, and may provide clues to the characteristics of the primitive translation system and how the translation apparatus evolved. For example, it would be interesting to examine prebiotic metals[41] in the iVTtRNA-based system. Elucidating how such metals affect tRNA function and the behavior of the overall translation machinery might be insightful.

In addition to the modifications present in the anticodon loop of tRNAs, which are directly involved in codon-anticodon interactions in the decoding process, many modified nucleotides are added to tRNAs, mainly for stabilization of the tertiary structure[42]. Therefore, a lack of all modification in iVTtRNAs might cause overall structural instability in reaction mixtures, and a shift in the optimal temperature may be needed to maximize the yield of synthesized proteins (Fig. 2d, e). As shown by our experiments using native tRNA mixtures, the *E. coli* translation apparatus is optimal at 37°C, but the optimal temperature was 30 °C when iVTtRNAs were employed, suggesting that a lack of modified nucleotides outside the anticodon loop may be one of the factors responsible for the low translation efficiency with iVTtRNAs, and this may reflect structural instability.

Our results further indicate the importance of modifications in the anticodon loop region for both translation efficiency and fidelity. The overall efficiency of protein synthesis was much lower when using iVTtRNAs than when using native tRNA mixtures (Supplementary Fig. 11), and this was not improved greatly by the addition of all iVTtRNAs, but it was improved by the addition of four specific iVTtRNAs (iVTtRNA$^{Ile}_{GAU}$, iVTtRNA$^{Pro}_{GGG}$, iVTtRNA$^{Asn}_{GUU}$, and iVTtRNA$^{Glu}_{CUC}$) (Fig. 3a). Surprisingly, addition of these iVTtRNAs also resulted in the recovery of translational fidelity, as shown by analysis of the specific activities of the synthesized proteins (Fig. 3b), suggesting that these four iVTtRNAs are more inefficient during aminoacylation, and presumably during decoding, than the other iVTtRNAs.

This was further validated by the introduction of modified nucleotides into these four iVTtRNAs. The introduction of t$^6$A37 into iVTtRNA$^{Ile}_{GAU}$ was most effective for the aminoacylation, the protein yield, and the fidelity (Fig. 5). The introduction of mnm$^5$U34 into iVTtRNA$^{Glu}_{UUC}$ was also shown to increase the aminoacylation and the protein yield (Fig. 5). The yield reached almost 40% of those with native tRNA mixtures (Fig. 5). Further attempts to modify the anticodon loop region, especially for those with U at 34 position may increase the number of iVTtRNAs and variety of codon sets that can be used for the reconstitution, which may facilitate the study of the effects on horizontal gene transfer and evolution[10].

Also, the improvement of the yield may require universal modifications on TΨC-loop and D-loop of tRNA, which stabilizes the tertiary structure of tRNA[42]. In this way, development of in vitro reconstitution systems for tRNA modification processes and their integration with the iVTtRNA-based system is an important future perspective for improving the productivity and decoding accuracy, both of which are crucial for the reconstitution of self-replicable artificial cells using a bottom-up approach. For this goal, development of the tRNA transcription-coupled and/or tRNA modification-coupled protein synthesis system is also important perspective. This approach may require the in vitro synthesis of sufficient amount of the tRNAs for efficient protein synthesis, requiring high amount of transcription substrates. Therefore, integration of nonequilibrium system through the construction of artificial cells[11,43] might be effective for this direction.

We also demonstrated that the iVTtRNA-based system can easily redesign the genetic code table, by switching the Ala codon from GCN to the Ser UCG codon (Fig. 4). Although the redesigning of the genetic code table has already been studied, e.g., flexizyme studies for peptide production, the protein synthesis with comparable activity to those synthesized with native tRNA mixtures based on the canonical genetic code table (Figs. 3b and 4c), was demonstrated in this study (Fig. 4a). LC-MS analysis (Supplementary Fig. 13) clearly showed that the codon reassigning from

Ser to Ala was performed efficiently as intended. Thus, the iVTtRNA-based system could contribute to the field of protein engineering by facilitating genetic code expansion. Since the system sustains translational activity for long peptide synthesis, and because many codons lack their cognate tRNA, there are no potential competitor tRNAs when extra tRNAs are incorporated in the system. Combined with aaRS engineering or flexizyme technology[2,3], as well as ribosome engineering[44], introduction of multiple noncanonical amino acids into long polypeptides and proteins could be made possible by reassignment of sense codons in combination with amber codon suppression techniques. The system may also prove to be a useful platform for developing orthogonal aaRS and tRNA pairs for sense codon suppression via directed evolution.

It is also interesting to note that natural tRNA mixtures failed to synthesize active proteins when the redesigned genetic code table was applied (Fig. 4c). The products obtained using natural tRNA mixtures were slightly shifted in the SDS-PAGE analysis, suggesting that Ser was incorporated in Ala positions (Fig. 4b). This indicates that template DNA based on the redesigned genetic code table has dual coding sequences; active proteins can be expressed only using the iVTtRNA-based system, and products are inactive when the canonical genetic code is applied. This feature can be applied to the study of dangerous proteins, such as virulence factors from pathogenic cells or viruses. The release of strains carrying dangerous genes into the environment can be a risk through horizontal gene transfer to living cells with unknown mechanisms. However, if the genetic materials for such dangerous proteins are based on the redesigned genetic code table, they would only be expressed as inactive proteins based on the canonical genetic code, even if strains are released, thereby reducing the risk to the environment. Therefore, the present system could also serve as a platform for studying dangerous proteins without biosecurity issues[45].

## Methods

**Preparation of protein components for reconstitution**. Components of the *E. coli* translation apparatus, including translation factors and aaRS, were prepared as previously described[46]. Preparation of RNaseP components, M1 RNA, and C5 protein were also prepared as described previously[29]. We modified the protocol for the C5 protein by precipitating it with 80% saturated ammonium sulfate, and dissolving it by dialyzing against buffer A (50 mM sodium acetate pH 6.5, 5 mM EDTA, 0.25 M NaCl, and 7 mM 2-mercaptoethanol). The resulting precipitate was recovered by centrifugation and dissolved by dialyzing against buffer B comprising 50 mM HEPES-KOH pH 7.6, 100 mM NH$_4$Cl, 6 M urea, and 10 mM dithiothreitol (DTT). Dissolved proteins were applied onto a 5 mL HiTrap SP HP column (#17115101, GE Healthcare, USA), washed with buffer B containing 7 mM 2-mercaptoethanol as a substitute for 10 mM DTT, and eluted with a linear gradient from 100 mM to 2 M NH$_4$Cl in buffer B. Fractions containing C5 protein were analyzed by SDS-PAGE, recovered, dialyzed against buffer D (50 mM HEPES-KOH pH 7.6, 0.8 M NH$_4$Cl, 10 mM MgCl$_2$, 2 M urea, 7 mM 2-mercaptoethanol), then further dialyzed against buffer D without urea. Resultant solutions were concentrated using an Amicon Ultra 3 kDa (#UFC800324, Merck Millipore, USA) and dialyzed against buffer D without urea containing 50% glycerol. The purified C5 protein was stored at −30 °C. We note that we can share all of the plasmids on requests.

**Preparation of modification enzymes for iVTtRNAs**. Modification enzymes for iVTtRNAs were prepared as follows. *E. coli* genes of TsaB, TsaC, TsaD, TsaE, TrmD, GlyA, MnmC, MnmE, and GidA were amplified from *E. coli* A19 genome using appropriate primers (Supplementary Data 5). Amplified genes for TsaC, TsaD, TsaE, GlyA, MnmC, MnmE, and GidA were cloned into pET15b (#69661, Merck Millipore, USA) as small ubiquitin-like modifier (SUMO) protein-fusion proteins where His-tag, SUMO protein, and modification enzymes were tandemly arranged. Genes for TsaB and TrmD were cloned into pET15b as His-tag fusion protein. All genes were cloned with Gibson assembly technique (#E2611, New England Biolabs, USA). Resultant plasmids were transformed into an *E. coli* BL21 (DE3) strain and grown in 1 L LB medium to an OD$_{660}$ of 0.6–1.0 at 37 °C. Overexpression was induced by the addition of IPTG to a final concentration of 1 mM for TsaB, TsaC, TsaD, TsaE, and TrmD, or 0.1 mM for GlyA, MnmC, MnmE, and GidA. After 3 h of cultivation at 37 °C, cells were harvested. TsaB, TsaC, TsaE, TrmD, and MnmE-overexpressed cells were resuspended in 40 mL of

Lysis buffer (50 mM Hepes-KOH, pH 7.6, 1 M NH$_4$Cl, 10 mM MgCl$_2$, and 7 mM 2-mercaptoethanol) and disrupted by sonication. Resultant lysate was centrifuged and the supernatant was recovered and mixed with 5 mL of cOmplete His-tag Purification Resin (#05893801001, Roche, Switzerland) for 1 h with rotator. The resin was washed with 100 mL of Lysis buffer and then the protein was eluted with 25 mL of Elution buffer (50 mM Hepes-KOH, pH 7.6, 400 mM KCl, 10 mM MgCl$_2$, 400 mM imidazole, and 7 mM 2-mercaptoethanol). TsaB and TrmD were concentrated by Amicon Ultra 3 kDa (#UFC800324, Merck Millipore, USA) and then dialyzed against Stock buffer (50 mM Hepes-KOH, pH 7.6, 500 mM KCl, 10 mM MgCl$_2$, 7 mM 2-mercaptoethanol, and 30% glycerol). They were flash frozen with liquid nitrogen and stored at −80 °C. For TsaC, TsaE, and MnmE, Ulp1 (#12588018, Thermo Fischer Scientific, USA) was added to the recovered fractions to a final concentration of 23 μg/ml to remove the His-tagged SUMO protein. The fractions were dialyzed against Cleavage buffer (50 mM Hepes-KOH, pH 7.6, 100 mM KCl, and 7 mM 2-mercaptoethanol) overnight while the Cleavage buffer contained 500 mM KCl for MnmE preparation. The dialyzed samples were again mixed with 5 mL of cOmplete His-tag Purification Resin with rotator. Then the flow-through fractions that contain modification enzymes were collected. Recovered samples were concentrated by Amicon Ultra 3 kDa for TsaE or Amicon Ultra 10 kDa (#UFC901024, Merck Millipore, USA) for TsaC and MnmE. They were dialyzed against Stock buffer, flash frozen with liquid nitrogen, and stored at −80 °C. For GlyA preparation, all buffer contained 10% Glycerol and the other procedures were same as MnmE preparation. TsaD was found to be insoluble after sonication. Therefore, the protein was pelleted by centrifugation at 20,400 × g at 4 °C for 45 min and it was resuspended in Lysis buffer supplemented with 4% Triton X-100. The suspension was again pelleted by centrifugation at 20,400 × g for 45 min. The pellet was dissolved with Lysis buffer supplemented with 6 M urea. The other procedure was same as MnmE preparation except that all buffers contained 2 M urea. For MnmC preparation, all steps until removal of His-tagged SUMO protein were same as GlyA preparation. Because MnmC nonspecifically bound to the His-tag purification resin, the purification with anion-exchange chromatography was selected. The solution after Ulp1 treatment was dialyzed against IEX buffer (50 mM Hepes-KOH, pH 7.6, 100 mM KCl, 10 mM MgCl$_2$, and 7 mM 2-mercaptoethanol) and then they were applied onto a 5 mL HiTrap Q HP column (#17115401, GE Healthcare, USA). The column was washed with 25 mL of IEX buffer and then MnmC was eluted with a liner gradient from 100 mM to 1 M KCl in IEX buffer. Fractions containing MnmC were concentrated by Amicon Ultra 30 kDa (#UFC903024, Merck Millipore, USA), dialyzed against Stock buffer, flash frozen with liquid nitrogen, and stored at -80 °C. We note that we can share all of the plasmids on requests.

**Preparation of iVTtRNA**. Genes for each iVTtRNA (Supplementary Data 1) were cloned into the pGEMEX-1 vector (#P2211, Promega, USA) between *Xba*I and *Bam*HI restriction sites. Using the resultant plasmids as templates, DNA templates for in vitro transcription were PCR-amplified using a T7 promoter primer as forward primer and appropriate reverse primers for each iVTtRNA (Supplementary Data 1). Each reverse primer contained a 2′-methoxy modification at the second nucleotide from the 5′-terminus to prevent template-independent addition of an extra nucleotide[28]. Products were purified by phenol/chloroform/isoamyl alcohol (25:24:1) extraction, followed by ethanol precipitation, and precipitants were dissolved in water. Run-off transcription of precursor iVTtRNAs with 27 extra nucleotides introduced at the 5′-terminus (5′-GGGAGACCA-CAACGGTTTCCCTCTAGA-3′) was performed using the resultant DNA templates for 3 h at 37 °C in 20 mL reaction mixtures containing 30 nM T7 RNA polymerase, 1 mM each ATP, GTP, CTP, and UTP, 40 mM HEPES-KOH pH 7.6, 20 mM MgCl$_2$, 1.5 mM spermidine, 5 mM DTT, 20 μg PCR products, and 0.2 U/mL inorganic pyrophosphatase (#10108987001, Roche, Switzerland). RNase P components composed of C5 protein and M1 RNA were subsequently added to reaction mixtures at 150 nM for removal of the 27 extra nucleotides, and reactions were incubated for 1 h at 37 °C. Transcribed iVTtRNAs were then processed with acidic phenol extraction followed by chloroform/isoamyl alcohol (10:1) extraction, then purification by anion-exchange chromatography. Samples containing iVTtRNAs were loaded onto 10 mL HiTrap Q HP columns (#17115401, GE Healthcare, USA) and washed with Q buffer (20 mM HEPES-KOH pH 7.6 and 200 mM KCl). iVTtRNAs were eluted with a linear gradient from 200 mM to 1 M KCl in Q buffer. Fractions containing target iVTtRNAs, determined by urea-PAGE, were recovered by isopropanol precipitation, and precipitated iVTtRNAs were dissolved in water and stored at -80 °C until use. We note that we can share all of the plasmids on requests.

**Modification of iVTtRNA**. t$^6$A37 modification was performed in the reaction mixture containing 50 mM Hepes-KOH, pH 7.6, 300 mM KCl, 20 mM MgCl$_2$, 5 mM DTT, 50 mM NaHCO$_3$, 1 μM CaCl$_2$, 1 mM ATP, 1 mM Thr, 5 μM TsaC, 5 μM TsaB, 5 μM TsaD, 5 μM TsaE, 4 A$_{260}$ unit/mL tRNA$^{Ile}_{GAU}$ or tRNA$^{Asn}_{GUU}$ or tRNA$^{Phe}_{GAA}$. m$^1$G37 modification was performed in the reaction mixture containing 50 mM Hepes-KOH, pH 7.6, 200 mM KCl, 10 mM MgCl$_2$, 36.4 μM S-adenosyl methionine (SAM), 1.5 μM TrmD, 10 A$_{260}$ unit/mL tRNA$^{Pro}_{GGG}$ or tRNA$^{Phe}_{GAA}$. mnm$^5$U34 was performed in the reaction mixture containing 50 mM Hepes-KOH, pH 7.6, 150 mM KCl, 12.5 mM MgCl$_2$, 5 mM DTT, 500 μM FAD, 1 mM tetrahydrofolate, 4 mM GTP, 2.5 mM NADH, 0.2 mM pyridoxal-5′-

phosphate, 1 mM Ser, 1 mM Gly, 36.4 µM SAM, 0.1 unit/µL recombinant RNase inhibitor (#2313 A, TaKaRa, Japan), 10 µM GlyA, 3 µM GidA, 3 µM MnmE, 2.5 µM MnmC, 6 $A_{260}$ unit/mL $tRNA^{Glu}_{UUC}$ or $tRNA^{Glu}_{CUC}$. After incubation at 37 °C for 2 h for $t^6A37$ and $m^1G37$ modification or 4 h for $mnm^5U34$ modification, tRNAs were processed with acidic phenol extraction followed by chloroform/isoamyl alcohol (10:1) extraction and recovered by isopropanol precipitation. Precipitated tRNAs were dissolved in water and stored at −80 °C. For $mnm^5U34$ modification, the reaction was again performed with recovered tRNAs. They were processed with acidic phenol extraction followed by chloroform/isoamyl alcohol (10:1) extraction, desalted by MicroSpin G-25 Columns (#27532501, GE Healthcare, USA), and recovered by isopropanol precipitation. Precipitated tRNAs were dissolved in water and stored at −80 °C. To quantify the modification efficiency, 57.1 µM [$^{14}C$]Thr was used instead of 1 mM Thr and the mixture without TsaD was used as control for $t^6A37$ modification. 36.4 µM S-[methyl-$^{14}C$]-adenosyl-Methionine was used and the mixture without TrmD was used as control for $m^1G$ modification and without GidA was used as control for $mnm^5U34$ modification. After incubation at 37 °C, aliquots (10 µL) were withdrawn and spotted on Whatman 25 mm GF/C filter disks (#1822-025, GE Healthcare, USA). Filter disks were washed twice with 10% TCA, then ethanol, followed by measurement of radioactivity by a liquid scintillation counter. For the quantification of mnm5U modification, tRNAs were purified as above and then 0.02 $A_{260}$ unit was spotted on the filter disks instead.

**Aminoacylation.** Aminoacylation experiments were performed according to a previous report[47]. Reaction mixtures (10 µL) contained 100 mM HEPES-KOH pH 7.6, 15 mM $MgCl_2$, 40 mM KCl, 1 mM DTT, 4 mM ATP, 1 unit/µL recombinant RNase inhibitor (#2313 A, TaKaRa, Japan), 50 nM or 1.5 µM aaRS corresponding to each amino acid, 20 µM [$^{14}C$]-amino acid or 68 µM [$^{14}C$]Asn for asparagine aminoacylation, and 2 $A_{260}$ unit/mL native tRNA. For measuring methylation, a mixture of 0.6 µM [$^{35}S$]Met and 3.4 µM cold L-methionine was used instead. [$^{14}C$]Cys was obtained by reducing [1,2,1′,2′-$^{14}C$]-cystine with 50 mM DTT at 37 °C for 15 min. For measuring cysteinylation, the DTT concentration was 5 mM. When native tRNA mixtures were used, 40 $A_{260}$ unit/mL tRNA mixtures (#10109541001, Sigma-Aldrich, USA) were added instead. After incubation at 37 °C for 30 min, aliquots (8 µL) were withdrawn and spotted on Whatman 25 mm GF/C filter disks (#1822-025, GE Healthcare, USA). Filter disks were washed twice with 10% TCA, then with ethanol, followed by measurement of radioactivity by a liquid scintillation counter.

**Octapeptide synthesis with the PURE system.** DNA templates encoding octapeptides were PCR-amplified with appropriate primers (Supplementary Data 2) using the pURE1 vector (#PUREV001, BioComber, Japan) containing a T7 promoter sequence and the ribosome binding site (Shine-Dalgarno sequence) as a template. Amplified DNA templates were purified using a QIAquick PCR purification kit (#28104, QIAGEN, Germany). Octapeptide synthesis reactions contained 50 mM HEPES-KOH pH 7.6, 100 mM potassium glutamate, 13 mM magnesium acetate, 2 mM spermidine, 1 mM DTT, 2 mM ATP, 2 mM GTP, 1 mM UTP, 1 mM CTP, 20 mM creatine phosphate, 10 µg/mL 10-formyl-5,6,7,8-tetrahydrofolic acid, 0.2 µM ribosome, 4 nM DNA template, proteinaceous PURE system components including translation factors and enzymes, amino acids, and tRNAs. The concentrations of proteinaceous PURE system components were as described in a previous protocol[46]. We note that all 20 aaRSs were included in this experiment, regardless of DNA templates and test codons. The composition and concentration of amino acids, which depend on the test codons, are listed in Supplementary Data 2. The composition of iVTtRNAs depends on the template, and reactions were performed using basal iVTtRNAs (Supplementary Data 2) in the presence or absence of test iVTtRNAs (Supplementary Data 2). The concentration of each iVTtRNA was fixed at 6 µM. When native tRNA mixtures were used, 56 $A_{260}$ unit/mL tRNA mixtures (#10109541001, Sigma-Aldrich, USA) were added instead. Reactions were performed for 60 min at 37 °C and aliquots were withdrawn, spotted onto Whatman 3 MM filter papers (#1822-025, GE Healthcare, USA), boiled in 10% TCA at 90 °C for 30 min to deacylate aminoacyl-RNAs. The radioactivity in the 10% TCA-insoluble fraction was measured with a liquid scintillation counter.

**Protein synthesis with the PURE system.** Protein synthesis experiments were performed with a PUREfrex 2.0 kit (#PF201, GeneFrontier Corporation, Japan) without tRNA mixtures in Solution I (Buffer mix). We additionally added 0.2 µM [$^{35}S$]Met, 4 nM PCR-amplified DNA template for DHFR or sfGFP expression (Supplementary Data 3), and a specified amount of iVTtRNA mixture or native tRNA mixture. Reactions were performed at 30 or 37 °C for 12 h, and synthesized proteins were analyzed.

**Analysis of synthesized proteins.** Synthesized DHFR and sfGFP containing radioactive [$^{35}S$]Met were analyzed by 15% SDS-PAGE, and the gel image was visualized using a BAS-5000 bio-imaging analyzer (GE Healthcare, USA). Time-course analysis of sfGFP expression was performed by measuring sfGFP fluorescence every 3 min using a StepOne qRT-PCR system (#4376373, Applied Biosystems, USA). Fluorescence images of synthesized sfGFP in reaction mixtures were

obtained using an LAS-4000 instrument (GE Healthcare, USA). The activity of synthesized DHFR was measured as described previously[44]. Reaction mixtures (2 mL) contained 50 mM MES-KOH pH 7.0, 25 mM TRIS-HCl pH 7.0, 25 mM ethanolamine, 100 mM NaCl, 10 mM 2-mercaptoethanol, 0.1 mM EDTA, 100 µM dihydrofolic acid, and a 10 µL aliquot of the PURE reaction mixture, and they were incubated at 37 °C for 15 min. Next, 20 mM NADPH was added to a final concentration of 200 µM, and the decrease in absorbance at 340 nm was measured over a period of 10 min with a V-550 spectrophotometer (Jasco, Japan). One unit of DHFR was defined as the amount of enzyme required to process 1 µmol of dihydrofolic acid in 1 min at 37 °C.

**LC-MS analysis of synthesized proteins.** DHFR with FLAG sequence at its terminus (Supplementary Data 3) was synthesized with a PUREfrex 2.0 kit (#PF201, GeneFrontier Corporation, Japan) without tRNA mixtures in Solution I (Buffer mix). We additionally added 4 nM PCR-amplified DNA template for DHFR tagged with FLAG sequence at its C-terminus (Supplementary Data 3) and a specified amount of iVTtRNA mixture or native tRNA mixture. Reactions were performed at 30 °C for 12 h. Synthesized DHFR was purified with Anti-FLAG M2 Magnetic Beads (#M8823, Sigma-Aldrich, USA). Aliquots (55 µL) of the reaction mixtures were mixed with 245 µL of the FLAG wash buffer (50 mM Hepes-KOH, pH 7.6, 150 mM NaCl, 20 mM $Mg(OAc)_2$), and further mixed with 30 µL beads for 1 h. The beads were washed with 210 µL of the FLAG wash buffer followed by washing with the buffer without $Mg(OAc)_2$ twice (210 µL and 180 µL). Then, DHFR was eluted with 50 µL of the FLAG wash buffer without $Mg(OAc)_2$ but supplemented with 100 µg/mL 3xFLAG peptide (#F4799, Sigma-aldrich, USA) after gently mixing for 1 h. Preparation of samples for LC-MS analysis was performed according to a phase transfer surfactant (PTS)-aided protocol[48]. The 4 µL of a dense PTS buffer (100 mM sodium deoxycholate, 100 mM sodium N-lauroylsarcosinate, and 500 mM $NH_4HCO_3$) was added to 40 µl of the eluted samples. They were reduced with 10 mM TCEP at 37 °C for 30 min, alkylated with 20 mM iodoacetamide at 37 °C for 30 min, and quenched with 20 mM Cys. Each reaction solution was divided into two portions. One was digested by 10 ng/µl Lys-C (#90051, Thermo Fisher Scientific, USA) and 10 ng/µl trypsin (#90057, Thermo Fisher Scientific, USA) at 37 °C overnight. The other was digested by 10 ng/µl Asp-N (#90053, Thermo Fisher Scientific, USA) at 37 °C overnight. After digestion, 10% trifluoroacetic acid (TFA) was added to a final concentration of 1%, and the reaction solutions were centrifuged at $15,000 \times g$ for 5 min to precipitate the detergents. Supernatants were desalted using self-prepared stage tips[49] and dried with SpeedVac. LC-MS analysis was performed using an Orbitrap mass spectrometer (LTQ Orbitrap Velos Pro, Thermo Fisher Scientific, USA) equipped with a nanospray ion source (Nanospray Flex, Thermo Fisher Scientific, USA) and a nano-LC system (UltiMate 3000, Thermo Fisher Scientific, USA). The dried peptide mixtures were dissolved in a solution containing 5% acetonitrile and 0.1% TFA, and each sample was applied to the nano-LC system. Peptides were concentrated using a trap column (#164535, 0.075 × 20 mm, 3 µm, Acclaim PepMap 100 C18, Thermo Fisher Scientific, USA) and then separated using a nano capillary column (#NTCC-360/100-3-153, 0.1 × 150 mm, 3 µm, C18, Nikkyo Technos, Japan) using two mobile phases A (0.1% formic acid) and B (acetonitrile and 0.1% formic acid) with a gradient (5% B for 5 min, 5–45% B in 45 min, 45-90% B in 1 min, and 90% B in 4 min) at a flow rate of 500 nL/min. Elution was directly electrosprayed (2.2 kV) into the MS (positive mode, scan range of 200–1500 m/z, 60,000 FWHM resolution)[50].

## Statistics and reproducibility
Error bars indicate standard deviation of triplicate measurements. Welch's $t$ test was applied to show statistical significance using values of triplicate measurements. Exact number of replicates and measurement values are shown in Supplementary Data 6.

**Reporting summary.** Further information on research design is available in the Nature Research Reporting Summary linked to this article.

## Data availability
DNA or amino acid sequences of the plasmids and DNA primers used in preparation of DNA templates for in vitro transcription are available in Supplementary Data 1, 2, 3, and 5. LC-MS data is available in Supplementary Data 4. Source data for main figures are presented in Supplementary Data 6.

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

## Acknowledgements

We thank Nono Takeuchi-Tomita for helpful discussions. KA was partly supported by an NIGMS grant (R35GM122560 to Prof. Dieter Söll in Yale University). This work was supported by a Grant-in-Aid to KH (19J12955), KA (15K16083), and YS (17H05680) from the Japan Society for the Promotion of Science (JSPS), the Human Frontier Science Program (RGP0043/2017 to YS), the Astrobiology Center Project of the National Institutes of Natural Sciences (AB271004 and AB281007 to KA, and AB311005 to YS and KA), and an intramural Grant-in-Aid from the RIKEN Center for Biosystems Dynamics Research (to YS).

## Author contributions

K.H., K.A., T.U., and Y.S. designed the study. K.H., K.A., and Y.S. wrote the manuscript. K.H., N.S., K.M., and Y.S. performed the experiments. S.A. and T.Y. prepared plasmids for tRNA transcription. All authors discussed the results and commented on the manuscript.

## Competing interests

The authors declare no competing interests.
