## [Peer Review File · Communications Biology]

This manuscript has been previously reviewed at another Nature Research journal. This document only contains reviewer comments and rebuttal letters for versions considered at Communications Biology.

REVIEWERS' COMMENTS:

Reviewer #1 (Remarks to the Author):

The reviewers have addressed my concerns.

Reviewer #2 (Remarks to the Author):

The manuscript by Hibi et al. reports the development of the PURE system containing only iVTtRNAs with an optimization that can be comparable with the cell-free protein synthesis system containing native tRNA mix. The authors demonstrated that iVTtRNA PURE system could easily reprogram genetic code, convincing its high potential for various applications. The revised manuscript has shown significant improvement by adequately addressing reviewers' comments. Its scientific contribution and significance were clearly described.

The reviewer has one minor question.

The choices of three octapeptides for three different sets of iVTtRNAs are not clear. Why and how did the authors choose such three peptides with different sequences?

Response to comments of Reviewer #2

The manuscript by Hibi et al. reports the development of the PURE system containing only iVTtRNAs with an optimization that can be comparable with the cell-free protein synthesis system containing native tRNA mix. The authors demonstrated that iVTtRNA PURE system could easily reprogram genetic code, convincing its high potential for various applications. The revised manuscript has shown significant improvement by adequately addressing reviewers' comments. Its scientific contribution and significance were clearly described.

The reviewer has one minor question.

The choices of three octapeptides for three different sets of iVTtRNAs are not clear. Why and how did the authors choose such three peptides with different sequences?

Our answer is as follows:

We thank the reviewer for the improvement of our manuscript. We have revised the corresponding places to describe why and how we chose three peptide sequences in detail (P. 6-7, lines 183-192).